# A Produce Prescription Program in Eastern North Carolina Results in Increased Voucher Redemption Rates and Increased Fruit and Vegetable Intake among Participants

**DOI:** 10.3390/nu14122431

**Published:** 2022-06-11

**Authors:** Mary Jane Lyonnais, Ann P. Rafferty, Susannah Spratt, Stephanie Jilcott Pitts

**Affiliations:** 1Albemarle Regional Health Services, Elizabeth City, NC 27909, USA; maryjane.lyonnais@arhs-nc.org (M.J.L.); annprafferty@gmail.com (A.P.R.); 2Department of Public Health, Brody School of Medicine, East Carolina University, Greenville, NC 27834, USA; spratts16@students.ecu.edu

**Keywords:** produce prescription program, fruit and vegetable intake, rural populations

## Abstract

Few produce prescription programs have taken place in rural areas, in the context of existing public health programs. Thus, the purpose of this mixed-methods study was to examine voucher redemption rates, change in fruit and vegetable intake, and suggestions for improvement among participants enrolled in a produce prescription program occurring in existing public health programs throughout rural eastern North Carolina. We examined voucher redemption rates and conducted pre- (*n* = 125) and post-intervention surveys assessing fruit and vegetable intake. *t*-tests were used to examine changes in intake pre- versus post-intervention among 50 participants. Participants (*n* = 32) also completed a semi-structured, telephone interview. Qualitative data were thematically analyzed to determine potential improvements. The overall voucher redemption rate was 52%. There was a 0.29 (standard deviation = 0.91, *p* = 0.031) cup increase in self-reported fruit intake comparing post- to pre-intervention data. Qualitative analyses indicated that participants enjoyed the financial benefits of the program and wanted it to continue. The produce prescription program was successful in increasing self-reported fruit intake among participants. More research is needed to determine if changes in intake persist when measured objectively, and on best methods for the program’s financial sustainability.

## 1. Introduction

Although fruit and vegetable (FV) intake is critical for optimal health [1,2,3], few Americans consume recommended amounts of FVs [4]. Produce prescription (Rx) programs, wherein health care providers or public health educators provide prescriptions or vouchers for redemption of FVs, are one way to address this public health problem. Prior studies have shown that produce Rx programs can help patients eat healthier [5,6,7], improve weight status [8], and reduce hemoglobin A_1_C (HbA_1_C) among diabetics [5,9,10].

The produce Rx program impacts on dietary intake are favorable: For example, Xie and colleagues [7] found that their produce Rx program was associated with increased healthy food purchasing among 699 participants with diabetes in Durham, North Carolina. Trapl, et al. [6] found a significant increase in FV intake among 224 participants enrolled in a produce Rx program in Ohio. Ferrer, et al. [5] found that participants (*n* = 29 in control group and 29 in the intervention group) who received regular produce boxes and nutrition education demonstrated improved diet quality scores. Saxe–Custack, et al. [11] found improvements in food security and diet after exposure to a pediatric produce Rx program among 122 participants in Michigan. Despite these findings, the produce Rx program impacts on health outcomes have been mixed. Cavanaugh and colleagues [8] found significant pre–post intervention differences in body mass index (BMI) among veggie Rx participants in upstate New York. Bryce and colleagues [9] found that the fresh Rx program in Detroit resulted in a statistically significant decrease in HbA_1_C but no changes in blood pressure (BP) or BMI. Ferrer, et al. [5], Veldheer, et al. [10], and Haddad–Lacle, et al. [12] all found reductions in HbA_1_C. Overall, in a scoping review of 23 food prescription programs, Little et al. [13] found that food prescriptions can improve FV intake and reduce food insecurity, but there was limited and mixed evidence for impacts on diet-related health outcomes (BP, weight, BMI, HbA1c) [13]. 

Qualitative studies of produce Rx programs have demonstrated that tailoring educational materials to seasonal produce available is important [14] as are positive experiences with providers and clinicians [14,15]. Schlosser, et al. [16] found that limited access to transportation, and limited and unstable incomes were economic barriers to program engagement and sustainability of health behavior change. However, there is scant literature on programs that occur in multiple counties in rural areas, with iterative improvements to the program. Little is known about how to improve rural programs to reach more individuals through existing public health programming. 

We previously conducted an evaluation of a pilot produce Rx program [17] in 2020, which had an 18% voucher redemption rate, and revealed that suggested improvements included starting the program earlier in the growing season and making efforts to increase awareness of the program and build relationships among community members and produce retailers. These improvements were made, and thus, the purpose of this study that used data from 2021 was to re-examine (1) voucher redemption rates, (2) change in FV intake among participants, and (3) to determine if the use of federal food assistance programs was associated with lower voucher redemption rates. We also conducted a qualitative analysis of additional ways we could improve the program in the future.

## 2. Materials and Methods

The Albemarle Regional Health Services (ARHS, Elizabeth City, NC, USA) Partnerships to Improve Community Health (PICH) Produce Rx program: 

The PICH Produce Rx Program partnered with cooperative extension programs, local health departments, federally qualified health centers, faith-based organizations, and a local hospital to provide produce “prescriptions” or vouchers to participants [17]. In brief, participants in nine counties were recruited from healthy lifestyle programs, nutrition education sessions, diabetes prevention programs, and routine healthcare visits after which they would be given a series of $5 vouchers, at least $20 total, to redeem for fresh FV from participating local farmers’ markets, grocers, and food stands. The nine counties were selected due to their being priority areas for the funder due to low socioeconomic status and they included public health programs that were willing to partner with the produce Rx program. As described in our previous report [17], each individual public health program recruited and enrolled participants based on its own eligibility criteria. The program length varied by county and specific program. Some participants were given vouchers one time and some were given vouchers several times during the season. In order to evaluate the program, participants completed a pre-intervention survey before receiving vouchers and were asked to complete a post-intervention questionnaire on the back of the voucher, after redeeming the vouchers. Participant identification numbers were assigned using the participant’s birthdate and initials. A sub-sample of participants also completed a qualitative interview in the fall of 2021. This study was reviewed and approved by the East Carolina University Medical Center Institutional Review Board, UMCIRB 20-001075.

### 2.1. Quantitative Study Methods

Participant recruitment and voucher redemption rates:

To determine voucher redemption rates, the ARHS Healthy Foods Coordinator maintained a spreadsheet that tracked the number of vouchers given out and those redeemed by the county and by the agency within the county. The number of vouchers redeemed was then divided by the number of vouchers distributed in each county to calculate the total redemption rate. The agency that distributed the vouchers was noted on the back of each voucher card so that the number of vouchers redeemed per program could be calculated. In some cases, the agency that distributed the voucher was not included on the voucher, but the voucher was redeemed at a farmers’ market in a specific county. In that case, the county where the voucher was redeemed was known, but the agency was unknown.

In both the pre- and post-intervention questionnaires, FV intake was assessed by using two questions developed by Townsend et al. to assess daily fruit and vegetable intake [18]. The response categories for each of these questions ranged from 0 to more than 3.0 cups/day and included pictures of cups of FV to aid in recall. In addition, as in our prior evaluation [17] questions were asked about barriers to eating fresh fruits and vegetables, food shopping venues, farmers’ markets, food insecurity [19], socio-economic and demographic characteristics, and the type of education they received from the participating program. The pre-intervention questionnaire was administered prior to voucher distribution in a variety of ways, i.e., self-administered on paper or online, or administered by an interviewer or participating program staff in person or by phone.

The post-intervention questionnaire was printed on the back of the vouchers and was self-administered. If multiple post-surveys were completed, only the voucher with the most recent date was used. If more than one voucher was redeemed on the same date, the average (or most frequently reported response) was used for each question. In addition to the FV intake questions, two other questions were included in the post-intervention questionnaire: one question about whether the participant had tried any new farmers’ markets because of the produce prescription initiative and a three-statement agree–disagree matrix about shopping at farmers’ markets. 

### 2.2. Data Analyses 

Data analyses were conducted by using SPSS for Windows, Version 25.0 (IBM Corporation, Armonk, NY, USA). Pre and post survey data were merged at the participant level by participant ID (combination of first and last initials and date of birth). For the post-survey data, only responses from the most recently used voucher were included in the merged dataset. If there were multiple vouchers redeemed on the same day, an average or the most frequently reported responses were used. Differences in FV intake between post-intervention minus pre-intervention were calculated, as was the total combined FV intakes for the pre- and post-intervention periods, respectively. Responses to the two food insecurity questions were combined to create an indicator of food insecurity that was defined as often or sometimes worried about or had run out of food during the past 12 months. Frequencies and proportions, and means and standard deviations when appropriate, were calculated for all variables. 

Statistical significance of change in FV intakes was examined by using paired *t*-tests comparing post- and pre-intervention intake among participants. A Spearman correlation analysis was used to examine the relationship between dose (number or proportion of vouchers redeemed) and reported FV intake. To determine if use of federal food assistance programs was associated with lower redemption rates, we conducted a *t*-test to compare mean proportion of vouchers redeemed by whether participants reported that they used SNAP and WIC. 

### 2.3. Qualitative Study Methods

#### 2.3.1. Participant Recruitment

Participants were asked in the pre-intervention quantitative survey if it would be okay to contact them again after they had finished the program, and if yes, they were asked to provide contact information including phone number, email address, and mailing address. A research associate trained in qualitative interview techniques called each pre-intervention survey participant who provided a phone number and asked if they would be willing to be interviewed over the phone regarding their experiences in the produce Rx program. If they agreed, they were interviewed at that time or called back at a more convenient time. Thirty-two participants completed a qualitative interview.

#### 2.3.2. Interview Guide

The interview guide included questions on how participants heard about the program, the type of program they participated in, how they received the vouchers, the number of vouchers they redeemed, the location of voucher redemption, the experience at the voucher redemption location, the FV items obtained with the vouchers, how the FV items were prepared, any changes in diet or health status the participant attributed to the program, and overall suggested improvements for the program. 

### 2.4. Qualitative Analyses

Interviews were audio recorded and transcribed verbatim by using Rev. Two researchers (SS and SBJP) each read 3 data-rich transcripts and created independent codebooks. The two researchers met to discuss their codebooks and created one consensus codebook. Then, each researcher independently coded each transcript, and met to discuss coding discrepancies and reach consensus on the final coding decisions for each transcript. Final themes were decided based upon how frequently each code was mentioned and the depth of discussion around each code. Once all transcripts were double-coded, one of the researchers coded all transcripts in NVivo (QSR, Melbourne, Australia) by using the consensus codes. Code summaries and reports were retrieved from NVivo and analyzed to find themes and subthemes from the coded transcripts.

## 3. Results

Table 1 shows voucher redemption rates for each program, county, and for all counties as a whole. There was a total of 2388 vouchers distributed. In general, there was wide variation in the number of vouchers distributed in each county, and the highest redemption rates were among county cooperative extension educational programs. 

One hundred thirty-nine participants completed a pre-intervention survey. Three duplicate cases were deleted, and an additional 11 cases were deleted because of missing data, leaving a working sample size of 125 cases (participants). There were initially 1578 records in the post-intervention survey dataset; 917 records were deleted due to no ID information (no first initial, no last initial, and no date of birth) or no responses to the post-intervention survey questions. An additional 491 duplicate records were removed (multiple voucher redemptions by the same participant), leaving one record per ID. There was a working sample size of 170 in the post-intervention survey dataset; however, only 164 included FV consumption data. Pre- and post-intervention survey data were successfully merged for 50 cases (See Appendix A).

Demographic, socio-economic, and food-related characteristics of participants are provided in Table 2. Nearly half of participants (48.8%) were aged 65 years or older and the majority (83.1%) were female. Nearly three-quarters (72.1%) were Black, and over half had some college education or were a college graduate. Nearly four in ten (37.6%) used SNAP/EBT at the time and only 7.2% used WIC. Of note, over 85% of respondents were from one specific county, which is 26.7% Black or African American, 70.1% White, 20.0% persons living in poverty [20]; 100% of the population is considered rural [21], and 19% [22] of the population is food-insecure.

About one-third of participants (31.2%) reported that it was not hard for them to eat fresh fruits and vegetables. However, 48.0% said fresh fruits and vegetables were not available in the neighborhood, 20.8% said they do not have enough money to buy them, and 10.4% said they spoil too quickly. The majority (97.6%) purchased food for their family from supermarkets, grocery stores, supercenters, and warehouses, while 41.6% went to convenience and dollar stores for food, and 14.4% went to a food pantry or shelter. One third (33.6%) reported that they had gone to a farmers’ market for food purchases in the past month, and the majority (85.6%) had ever shopped at a farmers’ market. Less than half of participants (45.1%) were classified as food-insecure.

In Table 3, we present opinions and attitudes toward farmers’ markets from the pre-intervention survey. The majority (96.7%) agreed with the statement, “I am interested in shopping at farmers’ markets” while only 35.5% agreed with the statement “The prices of fruits and vegetables at farmers’ markets are low compared to grocery stores” and 41.9% agreed with the statement, “Farmers’ markets always have the fruits and vegetables I want.” Nearly one-quarter (22.8%) disagreed with the statement “It is easy for me to get to farmers’ markets.” Overall, 65% (*n* = 149) of those completing the post-intervention survey said they tried a new farmers’ market because of the produce Rx program.

The frequencies of reported FV among the 125 participants who completed the pre-intervention questionnaire and the 164 participants who completed the post-intervention questionnaire were generally similar (Table 4). However, mean FV consumption at post-intervention was about twice that at pre-intervention. 

For 50 out of the 125 participants who completed the pre-intervention questionnaire, we were able to merge pre- and post-intervention data and to analyze the data for the change in FV intake. There was a 0.29 (standard deviation = 0.91, *p* = 0.031) cup increase in self-reported fruit intake comparing post- to pre-intervention data, and a 0.17 (standard deviation = 0.92, *p* = 0.210) cup increase in self-reported vegetable intake comparing post- to pre-study data. When reported FV intake was combined, the increase in combined FV intake was also statistically significant (mean 0.46 cups, standard deviation = 1.58, *p* = 0.047). However, when the dose was examined by using Spearman’s rho, neither number of vouchers redeemed nor the proportion of vouchers redeemed were correlated with a change in fruit (number of vouchers *p* = 0.727, proportion of vouchers *p* = 0.772) or vegetable consumption (number of vouchers *p* = 0.847, proportion of vouchers *p* = 0.825). 

Those participants who also used SNAP redeemed a lower proportion of their vouchers (mean = 0.375, SD = 0.240) compared with those who did not use SNAP (mean 0.543, SD = 0.305, *p* = 0.0436). In a regression model, SNAP was still statistically significant (*p* = 0.019) after adjusting for food insecurity, age, and race. In other words, those who receive SNAP benefits redeemed a lower proportion of their vouchers, on average, than those who did not report receiving SNAP benefits. However, the proportion of vouchers redeemed by whether the participants used WIC did not differ (*p* = 0.478). 

### Qualitative Interviews

The response rate for the participant interviews was 24.4%. Participants were called via phone, and there were many participants who were not reached because they either did not answer the call or return the call after a detailed voicemail was left (*n* = 69), or there was no way to leave a voicemail (*n* = 11). Additionally, there were several instances in which calls could not be completed, due to poor reception or lack of phone service entirely (*n* = 7). Finally, there were several participants who were reached but refused to participate in the qualitative interview process (*n* = 12).

Most people heard about the program through their family (*n* = 10) or friends (*n* = 8). There were five participants who reported learning about the program through the cooperative extension and four participants who learned about the program through a community recruiter. The remaining participants heard about the program through their physician (*n* = 2), coworkers (*n* = 2), or church (*n* = 1). Nutrition education for the program was mostly received over the phone (*n* = 11), with several participants attending in-person sessions (*n* = 6), attending online sessions through Zoom (*n* = 3), their physician (*n* = 1), or other sources (*n* = 1), and other participants reporting that they did not receive any nutrition education (*n* = 2). (Some respondents did not definitively state how they received nutrition education.) Most participants reported using all of their vouchers (*n* = 25), with others reporting using only some of their vouchers (*n* = 5), and two participants reported using none of their vouchers. Several participants who did not use all their vouchers reported the barrier to voucher use being that the available locations for voucher redemption did not have the food items that participants were looking for (*n* = 6). Most participants reported seeing a change in their FV intake (*n* = 24), with others reporting no change in their FV intake (*n* = 3). When asked to suggest improvements for the program, most participants reported that they had no suggestions to improve the program because it was already effective. However, some participants reported wanting more classes to be offered to get more vouchers (*n* = 3), wanting the program to run longer (*n* = 2), wanting more locations at which to shop (*n* = 2), higher voucher amounts (*n* = 2), and to increase promotion of the program (*n* = 3). Only three participants mentioned gardens, mostly in the context of reflecting on previous gardening experiences when younger or on how FV obtained using the vouchers were those not grown in the garden. Table 5 provides qualitative codes, operational definitions, number of times each code was referenced in total, and number of transcripts in which the code is referenced. Additionally, there are key illustrative quotes from participants highlighting aspects of the program that contributed to their overall feelings towards the program.

## 4. Discussion

We found a 51% voucher redemption rate in the PICH Produce Rx Program in 2021, which is a significant increase over the 18% redemption rate in the first year of the program (2020) [17]. The voucher redemption rate increase is likely due to increased community involvement, particularly in one community located in a food desert where members of a faith community and cooperative extension staff conducted community member recruitment and outreach. Overall, it is noteworthy that many of the highest redemption rates occurred through cooperative extension educational programs, suggesting that cooperative extension may be a good partner in future produce Rx programming. Our study, like others [5,6,7], found improvements in self-reported dietary intake among produce Rx participants. In the current study, among those with pre- and post-intervention data, there an increase of 0.29 cups for self-reported fruit intake and a 0.17 cup increase in self-reported vegetable intake comparing post- to pre-intervention data. This aligns with findings from Trapl, et al. [6] who found daily fruit intake increased from an average of 1.6 servings to 2.4 servings, and daily vegetable intake increased from 1.7 to 2.5 servings, comparing pre- to post-intervention. In a scoping review [13], it was found that although food prescriptions can improve FV intake and reduce food insecurity, there is a need for more rigorous studies, with larger sample sizes, control groups, and validated measures of diet, food security, and health outcomes. 

The current study built on the findings from our previous evaluation [17] to improve program implementation in the second year of the program, including beginning earlier in the growing season and making efforts to increase awareness of the program among community members. Sundberg, et al. [23] also worked with the community to create the Navajo FV Rx Program, including three cycles of feedback and improvements on the program. For instance, their program changed so that instead of families having to redeem at one specific store, all participating FV Rx retailers could receive the vouchers. Also, there was feedback that some stores lacked adequate FV selection, and so FV retailers had to have minimum produce stocking as a requirement for participation [23]. This study indicates the need to work with the community to improve implementation of similar produce Rx programs.

We found that those receiving SNAP benefits redeemed a lower proportion of their vouchers, on average, than those who did not report receiving SNAP benefits. The finding of the current study is in disagreement with findings from prior studies [10,24] wherein redemption rates were high among those using SNAP benefits. Future research should seek to understand how individuals are using produce prescriptions in combination with their SNAP benefits.

Approximately 23% of the sample disagreed with a statement that it was easy for them to get to farmers’ markets. This highlights a potential barrier to redeeming FV vouchers—difficult geographic access to the market. Both geographic and financial access to farmers’ markets are important for participants to procure and consume fresh FV. Shopping at farmers’ markets is associated with greater FV intake [25,26] and is an established strategy to promote FV intake. Future research should examine strategies that alleviate both financial and geographic barriers to accessing farmers’ markets through produce prescription programs. 

Our current study is limited in that changes in FV intake were measured by self-report, and we did not include any anthropometric or medical data such as BMI or BP. A further limitation is that surveys suffer from social desirability bias [27]. In addition, post-intervention survey data were matched with pre-intervention data for less than 50% of the original cohort. However, we did use a FV intake measure that has been used in prior studies. Furthermore, the majority of surveys were completed by individuals from just one of the counties, due to a high level of community mobilization. Educators in other counties had too many competing priorities to facilitate collecting pre-intervention survey data. Moving forward, additional efforts will be made to ensure the pre-intervention and follow-up surveys are completed by all participants. 

The study team is currently working with a mobile application developer to administer the program via a mobile application to ensure better follow-up and data collection throughout the evaluation. Future research will include exploration of the mobile application as a way to generate program income (e.g., via advertising) to sustain the program after grant funds are depleted. Future researchers should examine the effectiveness of the produce Rx program on objectively assessed FV intake by using tools such as the Veggie Meter ^®^, which has been used in prior studies to determine the effectiveness of public health nutrition interventions [28,29].

## 5. Conclusions

The produce prescription program was successful in increasing self-reported fruit intake among participants. More research is needed to determine if changes in intake persist when measured objectively, and on best methods to improve the program’s financial sustainability.

## Figures and Tables

**Table 1 nutrients-14-02431-t001:** Redemption rates for the PICH Produce Prescription Program in nine northeastern North Carolina Counties. Total redemption rates for each county are in bold font.

County A	Community Program	Number of Vouchers Provided	Redemption Rate
	County A Health Department (Diabetes Prevention Program)	68	Unknown
	County A Federally Qualified Health Center	122	Unknown
	County A Food Distribution	120	51.67%
	County A Church	120	Unknown
	**Total**		**43.95%**
**County B**			
	Cooperative Extension Program A	328	70.73%
	Cooperative Extension Program B	60	93.33%
	County B County Health Department	36	30.56%
	County B Department of Social Services	200	64.00%
	County B Federally Qualified Health Center	400	26.75%
	**Total**		**52.15%**
**County C**			
	County C Hospital Outpatient Clinic	200	13.50%
	**Total**		**13.50%**
**County D**			
	Local Garden Project	200	13.00%
	**Total**		**13.00%**
**County E**			
	Cooperative Extension Program C	240	Unknown
	Child-related non-profit/Housing Authority	40	Unknown
	**Total**		**45.00%**
**County F**			
	County F Health Department (WIC)	112	29.46%
	**Total**		**29.46%**
**County G**			
	Community Health Center	200	21.00%
	Cooperative Extension Program D	8	0.00%
	**Total**		**20.19%**
**County H**			
	County H Health Department (WIC)	65	28.51%
	Cooperative Extension Program H	80	46.51%
	Cooperative Extension/ Faith Based Organization	1166	79.64%
	**Total**		**73.50%**
**County I**			
	County I Health Department	5	17.86%
	County I Food Distribution	36	12.00%
	**Total**		**12.50%**
	TOTAL	**2388**	**51.40%**

**Table 2 nutrients-14-02431-t002:** Demographic and food-related characteristics of the participant sample, 2021 produce prescription program, *n* = 125.

Demographic Characteristics	*n* (%)
Age in years	
20–44	20 (16.3)
45-64	43 (35.0)
≥65	60 (48.8)
Female	103 (83.1)
Race-ethnicity	
Black	88 (72.1)
White	27 (22.1)
Hispanic	7 (5.7)
Education	
<High school graduate	11 (8.8)
High school graduate or GED	44 (35.2)
Some college	33 (26.4)
College graduate	37 (29.6)
County of Mailing Address	
County A	9 (7.6)
County B	6 (5.1)
County C	0
County D	0
County E	4 (3.3)
County F	0
County G	1 (0.8)
County H	96 (81.4)
County I	1 (0.8)
County J	1 (0.8)
Currently use SNAP/EBT	47 (37.6)
Currently use WIC	9 (7.2)
Number of people in household (including participant)	
1 person	37 (29.8)
2 persons	51 (41.1)
3 persons	17 (13.7)
>3 people	19 (15.3)
**Barriers to fruit and vegetable consumption**	
What makes it hard for you to eat fresh fruits and vegetables? (multiple responses allowed)	
They are not available in my neighborhood	60 (48.0)
I do not have enough money to buy them	26 (20.8)
I do not have enough space to store them	6 (4.8)
I do not have enough time to cook/prepare them	12 (9.6)
I do not have knowledge on how to cook produce	1 (0.8)
They spoil too quickly	13 (10.4)
I have health-related dietary restrictions or dental problems	6 (4.8)
Transportation is an issue for me	10 (8.0)
Other	8 (6.4)
It is NOT HARD for me to eat fresh fruits and vegetables	39 (31.2)
**Food shopping venues**	
Thinking about the past month, what types of stores and markets have you or your family gone to for food? (multiple responses allowed)	
Supermarket, grocery store, supercenter, or warehouse	122 (97.6)
Convenience, dollar variety, or corner store	52 (41.6)
Farmers’ market	42 (33.6)
Food pantry or shelter	18 (14.4)
Other	11 (8.8)
Have ever shopped at a farmers’ market	107 (85.6)
**Indicators of food insecurity**	
Within the past 12 months we worried whether our food would run out before we got money to buy more.	
Often true	5 (4.1)
Sometimes true	39 (32.0)
Never true	78 (63.9)
Within the past 12 months the food we bought just didn’t last and we didn’t have money to get more.	
Often true	6 (4.8)
Sometimes true	42 (33.9)
Never true	76 (61.3)
Don’t know or refused to answer	
Classified as food-insecure (either often or sometimes worries about or runs out of food)	55 (45.1)

**Table 3 nutrients-14-02431-t003:** Pre-intervention (*n* = 125). and post-intervention (*n* = 165) attitudes about farmers’ markets (*n* = 125).

Pre-Attitude Attitude Statements	Agree *n* (%)	Neither Agree nor Disagree *n* (%)	Disagree *n* (%)
I am interested in shopping at farmers’ markets.	119 (96.7)	4 (3.3)	0 (0.0)
It is easy for me to get to farmers’ markets.	74 (60.2)	21 (17.1)	28 (22.8)
I feel welcome at farmers’ markets.	80 (65.6)	41 (33.6)	1 (0.8)
The staff and vendors at farmers’ markets are easy to talk to.	79 (63.7)	44 (35.5)	1 (0.8)
Farmers’ markets sell good quality, fresh food.	83 (66.9)	41 (33.1)	0 (0.0)
Farmers’ markets always have the fruits and vegetables I want.	52 (41.9)	64 (51.6)	8 (6.5)
The prices of fruits and vegetables at farmers’ markets are low compared to grocery stores.	44 (35.5)	73 (58.9)	7 (5.6)
Farmers’ markets are a good place to meet new people.	62 (50.0)	58 (46.8)	4 (3.2)
Post-Attitude Attitude Statements	Agree ***n*** (%)	Neither Agree nor Disagree ***n*** (%)	Disagree ***n*** (%)
I visit farmers’ markets more now than before the produce prescription program.	128 (79.5)	22 (13.7)	11 (6.8)
I will shop at farmers’ markets in the future.	149 (92.0)	12 (7.4)	1 (0.6)
Shopping at farmers’ markets has made it easy for me to include more fresh produce into my and my family’s diet.	149 (90.3)	15 (9.1)	1 (0.6)

**Table 4 nutrients-14-02431-t004:** Daily fruit and vegetable intake pre- and post-intervention: comparison of two points in time.

	Pre-Intervention (*n* = 125)	Post-Intervention (*n* = 164)
Daily fruit intake in past 7 days	*n* (%)	*n* (%)
None	8 (6.4)	3 (1.8)
0.5 cup	13 (10.4)	13 (7.9)
1.0 cup	31 (24.8)	42 (25.6)
1.5 cups	13 (10.4)	16 (9.8)
2.0 cups	27 (21.6)	39 (23.8)
2.5 cups	5 (4.0)	16 (9.8)
3.0 cups or more	28 (22.4)	35 (21.3)
Mean (std. dev.) fruit consumption (cups/day)	1.7 (0.94)	3.6 (1.72)
Daily vegetable intake in past 7 days		
None	1 (0.8)	2 (1.2)
0.5 cup	5 (4.0)	7 (4.3)
1.0 cup	28 (22.4)	33 (20.4)
1.5 cups	13 (10.4)	11 (6.8)
2.0 cups	31 (24.8)	44 (27.2)
2.5 cups	10 (8.0)	18 (11.1)
3.0 cups or more	37 (29.6)	47 (29.0)
Mean (std. dev.) vegetable consumption (cups/day)	2.0 (0.83)	4.0 (1.66)

**Table 5 nutrients-14-02431-t005:** Summary of participant interview transcript codes, definitions, and uses.

Code and Operational Definition	Illustrative Quote	Number of Times Code Was Used All Together	Number of Transcripts in Which Code Was Used
Financial Benefits - Participant discusses financial benefits of program	“Everything is so expensive now, so you just get the necessities. You might pick up a can of string beans versus getting fresh string beans or you might pick up a can of tomatoes instead of the fresh tomatoes. I think the fresh fruit and vegetables are always better. So I bypass the fruits when I’m spending my own money because they’re pricey.”- Participant 22	33	19
Change in FV intake-Participant discusses change or no change in FV intake and/or if the program helped them incorporate more fruits and vegetables.	“Yes ma’am. By getting [those] vouchers, by me learning about the vouchers, it made me eat more veggies because really I don’t like veggies. And by me getting [those] vouchers and I went and picked out the stuff out I like, it made me eat them and I really enjoyed it.”- Participant 15	41	27
Limited Store Availability- Participant mentions lack of places to shop for food	“And you know this little town…I don’t know if you know where [Rural Town A] is at. It ain’t really hardly nothing down here. You have to go so far to get to a store.” -Participant 15	18	11
How Vouchers Helped Use Skills- Participant mentions how having the vouchers helped them use skills they learned in the program	“For one, I got vegetables that I wouldn’t normally get to cook and I got to try them. So that helped out a lot.” -Participant 12	13	11
Confidence to Improve Health Habits- Participant describes how program increased confidence for healthier habits.	“Yeah, it has, those things that they taught. And it gave me more access to the vegetables with organic produce, so yeah, I was happy.” -Participant 10	12	11
Health Status Impacts- Participant mentions changes in health since starting program	“I work in the health field, so well, I’ve been out for a while, but just working in the health field and the impact of COVID, I think kind of incorporating my vegetables and stuff, it helped build my immunity a whole lot. I would say, even when I caught COVID, just eating more vegetables and incorporating more of what I needed in my body versus eating a whole lot of meat. I actually got to a point where I got tired of eating a lot of meat and I ate more vegetables.” -Participant 30	29	23
Continue Program- Participants discuss wanting the program to continue	“Yeah, I pray to God that the vouchers still continue going on because there’s lots of people in the community, really needs it. It’s good for all folks … and I hope the program stay in the county.” -Participant 20	13	10
Gardening- Participant mentions any aspect of gardening	“We normally raise a nice garden and we freeze, even at our age, we freeze stuff and we go out and pick fresh stuff. But when we got these vouchers, it was a lot easier to go to the store and buy the same thing I could have been raising.” -Participant 6	5	3
Nutrition Education Feedback- Participant discusses any education they received and/or if they were part of a nutrition program.	“Well, they showed me certain things where you could cook with olive oil, which I never used olive oil before. I used to use like the canola oil or vegetable oil. But now I’ve learned about olive oil.”-Participant 7	54	32

## Data Availability

Data can be obtained from the authors upon reasonable request and IRB approval.

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
