# Peer review of "A Produce Prescription Program in Eastern North Carolina Results in Increased Voucher Redemption Rates and Increased Fruit and Vegetable Intake among Participants"

_nutrients, 2022, doi:10.3390/nu14122431_

Round 1

Reviewer 1 Report

The authors provide a descriptive report of the impact of a multi-county produce prescription program (PPRx) and in doing so they contribute to the body of literature related to such programs. I think the manuscript could be strengthened by emphasizing the more unique aspects of the  program, as suggested below.

1) Introduction - Currently, the introduction provides a literature review of the PPRx on FV intake and health outcomes, but this doesn't build a rationale for why your manuscript is important (if there's already data on FV intake, what is important about your study?; health outcome data is equivocal, but your study doesn't look at those outcomes). Instead, I would focus on the unique strengths of your program-  the fact that this this is a broad program (multiple counties) in a rural area and the iterative improvements to the program as you went from a pilot to a large program. The literature review could be focused on what is known/ unknown in rural programs, and more details could be provided on the process of pilot, program improvements, and expansion (currently, mentioned briefly in the last paragraph, but should be elevated more).

2) Methods - I think the methods section needs a lot more information to understand the context and the program. 

  • Setting - please provide more details about the demographics, rural characteristics, food security / health disparities data for the region where the program took place.
  • Program details - more information is needed about the program itself - how were individuals recruited? Please describe the eligibility / selection criteria, and whether criteria varied by program/county or was it the same across the entire program. How long did the program last for a participant? how often were vouchers given? It is difficult to determine if this was a one-time voucher distribution per participant or if participants were enrolled longitudinally. If the former, the methods should describe whether there may be duplicates, i.e. an individual who received vouchers, completing pre/post and then at a later time, received another set of vouchers, completing pre/post again.   
  • Data collection - the method for tracking voucher redemption rates by agency is described, but in the results, the agency is unknown. Please describe why the agency may be unknown and how the county was then determined.
  • Data collection: It is unclear how voucher redemption was tracked individually; please add this information.
  • Food insecurity questions - please add citation for this 
  • The methods states that the post questionnaire was on the back of the voucher and the results section states multiple surveys may have been completed. If possible, please provide.

 RESULTS:

  • Please provide a flow chart summarizing the total # individuals enrolled in the program, how many completed pre only, post only, pre-post questionnaires.
  • Table 1 is interesting. In the narrative, it may be useful to flag the variation in terms of # vouchers dispensed.
  • Looking at the correlation between voucher redemption and Food Assistance Program (FAP) participation, please consider doing a multivariable analysis to control for potential confounders.
  • In the Methods section, the authors mention that several questions about farmers' markets were also collected at exit. Please present these findings, including pre-post (e.g. Table 3) if the same questions were asked.  
  • For qualitative methods, response rate was low. Please provide information about reasons (unable to reach, declined, etc). 
  • The gardening comment that was selected is interesting because it suggests there could be an impact of the PPRx on less gardening. Was this quote provided because it was representative? If not, it may be important to comment about the main findings related to gardening.

DISCUSSION

  • Some of the literature review in the introduction could be moved to the discussion, and changes in FV intake could be quantitatively compared with the study findings, with a particular emphasis on comparing with other rural programs.
  •  One of the original aims was to determine if voucher redemption correlated with participation in Food Assistance Program.  It would be important to mention the study finding, and provide an interpretation and /or ways the team has responded to these findings.
  • Acknowledgements:  The PPRx is called PICH - was the program supported by a CDC PICH grant? If so, please acknowledge. 
  • Overall: there are numerous typos; a thorough review for grammar / spelling is recommended 

Author Response

Author’s response: Thank you for your thorough review and comments. We have made many revisions to improve the paper based upon your review as outlined in the table below.

Reviewer 1 Comments

How addressed

1) Introduction - Currently, the introduction provides a literature review of the PPRx on FV intake and health outcomes, but this doesn't build a rationale for why your manuscript is important (if there's already data on FV intake, what is important about your study?; health outcome data is equivocal, but your study doesn't look at those outcomes). Instead, I would focus on the unique strengths of your program-  the fact that this this is a broad program (multiple counties) in a rural area and the iterative improvements to the program as you went from a pilot to a large program. The literature review could be focused on what is known/ unknown in rural programs, and more details could be provided on the process of pilot, program improvements, and expansion (currently, mentioned briefly in the last paragraph, but should be elevated more).

Thank you for this suggestion. We added more information to the introduction to focus on the unique strengths of the paper as suggested. For example, see lines 54-56.

2) Methods - I think the methods section needs a lot more information to understand the context and the program. Setting - please provide more details about the demographics, rural characteristics, food security / health disparities data for the region where the program took place.

We added more details on the one county where the majority of survey respondents live.

The demographic data for the sample is provided in Table 2 and summarized in the results.  See pages 5-7, and lines 166-168.

  • Program details - more information is needed about the program itself - how were individuals recruited? Please describe the eligibility / selection criteria, and whether criteria varied by program/county or was it the same across the entire program. How long did the program last for a participant? how often were vouchers given? It is difficult to determine if this was a one-time voucher distribution per participant or if participants were enrolled longitudinally. If the former, the methods should describe whether there may be duplicates, i.e. an individual who received vouchers, completing pre/post and then at a later time, received another set of vouchers, completing pre/post again.   

Each individual public health program recruited and enrolled participants based on its own eligibility criteria. The program length varied by county and specific program. Some participants were given vouchers one time and some were given vouchers over several times during the season. The analysis dataset was cleaned and duplicates were removed. We have added these details to the methods section, for example, lines 70-81.

If multiple post-surveys were completed, only the voucher with the most recent date was used. If more than one voucher was redeemed on the same date, the average (or most frequently reported response) was used for each question. (See line 112.)

  • Data collection - the method for tracking voucher redemption rates by agency is described, but in the results, the agency is unknown. Please describe why the agency may be unknown and how the county was then determined.

In some cases, the agency that distributed the voucher was not included on the voucher, but the voucher was redeemed at a farmers’ market in a specific county. In that case, the county where the voucher was redeemed was known, but the agency was unknown. We have added this to the paper, lines 89-91.

  • Data collection: It is unclear how voucher redemption was tracked individually; please add this information.

Participant identification numbers were assigned using the participant’s birthdate and initials. (Lines 110-112.)

  • Food insecurity questions - please add citation for this 

We have added the Hager, et al. reference for the food security questions. See line 97.

  • The methods states that the post questionnaire was on the back of the voucher and the results section states multiple surveys may have been completed. If possible, please provide.

 RESULTS:

  • Please provide a flow chart summarizing the total # individuals enrolled in the program, how many completed pre only, post only, pre-post questionnaires.

We have provided a flow chart to illustrate the number of individuals completing the various questionnaires.

  • Table 1 is interesting. In the narrative, it may be useful to flag the variation in terms of # vouchers dispensed.

We have added this information as follows: “In general, there was wide variation in the number of vouchers distributed in each county, and the highest redemption rates were among county cooperative extension educational programs.” See lines 150-151.

  • Looking at the correlation between voucher redemption and Food Assistance Program (FAP) participation, please consider doing a multivariable analysis to control for potential confounders.

We have added the results of the multivariable analyses to control for potential confounders. See lines 204-206.

  • In the Methods section, the authors mention that several questions about farmers' markets were also collected at exit. Please present these findings, including pre-post (e.g. Table 3) if the same questions were asked.  

These questions about farmers’ markets were not the same at pre-and post-data collection. However, we have added data from the post-surveys into Table 3, see page 7.

  • For qualitative methods, response rate was low. Please provide information about reasons (unable to reach, declined, etc). 

We have provided more details on reasons the response rate was low, see lines 211-215.

  • The gardening comment that was selected is interesting because it suggests there could be an impact of the PPRx on less gardening. Was this quote provided because it was representative? If not, it may be important to comment about the main findings related to gardening.

Only three participants mentioned gardening, so the reviewer is good to ask about representativeness – this quote was not representative. The other quotes are as follows:

“Yeah. Fresh vegetables in them. I grew up my dad before he passed, always had a huge garden. So you were used to just fresh vegetables, fresh tomatoes, everything from the garden. And my mom used to do a lot of canning, so I grew up with the vegetables and the ... My mom was like, "Nothing came from the box and nothing came from the can."

“So there was no limit to like what I really got as far as I got potatoes or onions, collared greens, and so, or different varieties of greens. Mostly I got a lot of vegetables. I'm, I'm allergic to some fresh fruits, so I didn't really buy, I really used it for vegetables that I didn't grow myself.”

So there was not an impact on gardening – We added the following, see lines 230 - 232: Only three participants mentioned gardens, mostly in the context of reflecting on previous gardening experiences when younger or on how FV obtained using the vouchers were those not grown in the garden.

  • Some of the literature review in the introduction could be moved to the discussion, and changes in FV intake could be quantitatively compared with the study findings, with a particular emphasis on comparing with other rural programs.

We have moved literature from the introduction into the discussion to compare findings with those of other studies. For example, see lines 246 – 249.

  •  One of the original aims was to determine if voucher redemption correlated with participation in Food Assistance Program.  It would be important to mention the study finding, and provide an interpretation and /or ways the team has responded to these findings.

We have mentioned our finding related to this aim and reflected on this finding in the discussion section. Please see lines 262-266.

Acknowledgements:  The PPRx is called PICH - was the program supported by a CDC PICH grant? If so, please acknowledge. 

The PICH title was left over from the original CDC PICH grant. In order to maintain community branding and continuity, the name stayed the same but it was no longer funded by CDC.

Overall: there are numerous typos; a thorough review for grammar / spelling is recommended.

We have made many changes throughout the paper to address the grammar and typos.

Reviewer 2 Report

The paper examines voucher redemption rates, measures self-reported fruit and vegetable intake, and suggests improvement among participants enrolled in a produce prescription program occurring in existing public health programs in rural eastern North Carolina. I think the research is interesting and the manuscript is well organized. However, the paper lacks clarity in the procedures of the data collection and does not bring forward the main contributions. Please see my comments below. I hope it contributes to guiding you through your revision.

Pg. 1, line 38: who instead of wo

Pg. 2, line 52: HbA1C were instead of HbA1Cwere

Pg. 2, line 63: income were instead of incomevwere

Pg. 2, lines 65-73: It is here where I miss the contribution of the research. The authors mention previous studies on dietary intake and health outcomes of Rx programs, however, how is this research different from the rest? What do you do differently? What do you add to the literature?

Pg. 2 – the section of Materials and Methods is confusing. Based on what criteria did you choose to include the 9 counties in your study (if any)? Based on what criteria did you decide on the number of vouchers (5$ min and 20$ max) given to participants?

Pg. 2, line 85: Could you give some more details on the questions and the procedures of the post-intervention questionnaire?  

Merging the section on quantitative and qualitative methods would avoid confusion. The same applies to both the data analysis sections. In addition, a table summarizing all the methods used in the study and the timing of each method would clarify a lot. What I am missing is the details of the pre-intervention process of the produce Rx program.

Pg. 5, lines 174-175: What was the main reason of asking this question? “Hard” in terms of taste, availability, price….etc.

Pg. 8, lines 212-214: Could you shortly elaborate on what that result means?

Pg. 8, lines 231-232: Besides self-reported measures, did you use any medical measures (e.g., changes in body mass index, etc.) to report results?

Pg. 8, lines 234-235: Were participants provided with vouchers based on the classes they would attend? If yes this has to be clearly stated in the methodological part of the research.  

Pg. 10, line 302: Please provide suggestions for future research. Additionally, surveys do also suffer from social desirability bias (Camerer and Lovalle, 1999), hence this is another limitation of your research.

Camerer, C., & Lovallo, D. ‘Overconfidence and Excess Entry: An Experimental Approach’, The American Economic Review, Vol. 89(1), (1999), pp. 306–318.

The conclusions are too short. What are the implications of this research? Can your results contribute to increasing the FV consumption if there will not be vouchers? What are the contributions of your work and who can use the results? 

Author Response

Author’s response: Thank you for your thorough review and comments. We have made many revisions to improve the paper based upon your review as outlined in the table below.

Reviewer Comment

Author’s response

Pg. 1, line 38: who instead of wo

We have made this change.

Pg. 2, line 52: HbA1C were instead of HbA1Cwere

We have deleted the section of text that included this typo to better frame the introduction and contribution of the research.

Pg. 2, line 63: income were instead of incomevwere

We have made this change.

Pg. 2, lines 65-73: It is here where I miss the contribution of the research. The authors mention previous studies on dietary intake and health outcomes of Rx programs, however, how is this research different from the rest? What do you do differently? What do you add to the literature?

The contribution of the research to the broader literature is now clearer in the introduction. We focus on the rural context and iterative nature of improvements to the program. See lines 54 – 56.

Pg. 2 – the section of Materials and Methods is confusing. Based on what criteria did you choose to include the 9 counties in your study (if any)? Based on what criteria did you decide on the number of vouchers (5$ min and 20$ max) given to participants?

Pg. 2, line 85: Could you give some more details on the questions and the procedures of the post-intervention questionnaire?  

The nine counties were selected as they are priority counties for the funding agency based upon overall low socioeconomic status or they included public health programs that were willing to work on the Produce Rx Project.. We have added this into the methods section. See lines 73-75.

We have provided more details on the questions and procedures of the post-intervention questionnaire, as well as results of the farmers’ market-related questions on the post-intervention questionnaire.

Merging the section on quantitative and qualitative methods would avoid confusion. The same applies to both the data analysis sections. In addition, a table summarizing all the methods used in the study and the timing of each method would clarify a lot. What I am missing is the details of the pre-intervention process of the produce Rx program.

We would like to keep these two sections separated as we feel it keeps the paper organized better. 

We have clarified the pre-intervention process within the methods section. See lines 75 – 81.

Pg. 5, lines 174-175: What was the main reason of asking this question? “Hard” in terms of taste, availability, price….etc.

We asked this question to learn about barriers to consuming FV so that they could be addressed in the future. The question was to learn what the main barriers to FV were among our participants.

Pg. 8, lines 212-214: Could you shortly elaborate on what that result means?

This means that those who receive SNAP benefits redeemed fewer vouchers, on average, than those who did not report receiving SNAP benefits. We have added this to the results section lines 204-206.

Pg. 8, lines 231-232: Besides self-reported measures, did you use any medical measures (e.g., changes in body mass index, etc.) to report results?

No – we did not use any medical measures such as changes in BMI. This would have been good information to have, though. We have added this as a limitation. (Lines 276-277)

Pg. 8, lines 234-235: Were participants provided with vouchers based on the classes they would attend? If yes this has to be clearly stated in the methodological part of the research.  

We have added details to the methods regarding how participants received vouchers, see lines 74-77.

Pg. 10, line 302: Please provide suggestions for future research. Additionally, surveys do also suffer from social desirability bias (Camerer and Lovalle, 1999), hence this is another limitation of your research.

Camerer, C., & Lovallo, D. ‘Overconfidence and Excess Entry: An Experimental Approach’, The American Economic Review, Vol. 89(1), (1999), pp. 306–318.

We have added suggestions for future research and have added the limitation of social desirability bias, lines 275-277.

The conclusions are too short. What are the implications of this research? Can your results contribute to increasing the FV consumption if there will not be vouchers? What are the contributions of your work and who can use the results? 

We have added two paragraphs to bolster the discussion section and discuss implications of the research.  See lines 262-272.

Round 2

Reviewer 2 Report

I would like to congratulate the authors for responding to all of my comments. I feel that the manuscript has improved not only in terms of presenting a clear contribution to the reader but also in terms of clarifying the procedures and methods used in the research. That being said, I do not have further comments.